# Rift Valley fever virus and *Coxiella burnetii* infections among febrile patients, Eastern Ethiopia

Dadi Marami[1]*, Adane Mihret[2], Nega Assefa[1], Alemseged Abdissa[2], Mahlet Osman[2], Gizachew Gemechu[2], Jacob S. Witherbee[3], Adargachew Mulu[2], Rea Tschopp[2,4,5¤]

1 College of Health and Medical Sciences, Haramaya University, Harar, Ethiopia, 2 Armauer Hansen Research Institute, Addis Ababa, Ethiopia, 3 Center for Disease Control and Prevention, Atlanta, Georgia, United States of America, 4 Swiss Tropical and Public Health Institute, Allschwil, Switzerland, 5 University of Basel, Basel, Switzerland

¤ Current address: Armauer Hansen Research Institute, Addis Ababa, Ethiopia
* dmarami4@gmail.com

## Abstract

### Background

Rift Valley fever (RVF) virus and *Coxiella burnetii* infections are significant public health concerns in East Africa, causing recurring outbreaks. However, the prevalence of these pathogens among febrile patients in Ethiopia remains unknown. This study aimed to determine the prevalence and associated factors of these infections among febrile patients.

### Methods

A multisite cross-sectional study was conducted among 415 randomly selected adult febrile patients from health facilities in Shinile and Dire Dawa, Ethiopia, between March 01, 2023, and February 28, 2024. Serum samples were tested for the presence of antibodies against RVF virus and *C. burnetii* infections using various Enzyme Linked Immunosorbent Assays. Polymerase Chain Reaction (PCR) was used to detect RVF virus RNA and *C. burnetii* DNA in blood samples. A multivariable logistic regression model was used to identify predictive factors. A *p* value <0.05 was considered statistically significant.

### Results

Of the 402 serum samples analyzed, 21 (5.2%) tested positive for immunoglobulin G (IgG) antibodies against RVF virus, and 86 (21.4%) tested positive for *C. burnetii* Phase I and Phase II antibodies. No RVF virus IgM was detected. Among the *C. burnetii* antibodies positive sera, 6 (7.0%) were positive for Phase II IgG antibodies. No blood samples tested positive for RVF virus RNA or *C. burnetii* DNA. Febrile patients

**Data availability statement:** The authors
confirm that all data underlying the findings are
fully available without restriction. All relevant
data are within the paper.

**Funding:** The author(s) received no specific
funding for this work.

**Competing interests:** The authors have
declared that no competing interests exist.

aged ≥35 years had significantly higher odds of RVF virus exposure (AOR: 3.1, 95% CI: 1.3-7.8). Females (AOR: 1.7, 95% CI: 1.1-2.9), rural residents (AOR: 2.4, 95% CI: 1.3-4.5), and febrile patients who disposed of dead animals (AOR: 2.6, 95% CI: 1.2-5.6) exhibited significantly higher odds of *C. burnetii* infection.

## Conclusions

This study reveals significant but underrecognized exposure to RVF virus (5.2%) and *C. burnetii* (21.4%) infections among febrile patients. Risk factors for RVF included older age, whereas *C. burnetii* infection was associated with females, rural residents, and exposure to dead animals. Health authorities are advised to consider these infections in the differential diagnosis of fever, implement active surveillance, and target public health interventions.

## Author summary

Rift Valley fever virus and *C. burnetii* are priority pathogens that cause severe febrile illnesses with overlapping symptoms, yet their burden in Ethiopia remains poorly understood. This study investigated the prevalence of these infections among 402 febrile patients in Eastern Ethiopia. The detected seroprevalence was 5.2% for RVF virus and 21.4% for *C. burnetii* infection. No RVF virus RNA or *C. burnetii* DNA was detected, which is likely attributable to the delayed healthcare-seeking behavior and low pathogen levels at presentation. Notably, none of the seropositive febrile patients were clinically suspected to have these infections before testing. Risk factor analysis revealed that older febrile patients (≥35 years) had significantly higher odds of RVF virus infection. Conversely, female sex, rural residency, and handling dead animals increased the odds of *C. burnetii* infection, suggesting that occupational exposure is a key risk factor. These findings suggest that the true burden of RVF virus and *C. burnetii* infections is underestimated in this region. Active surveillance and enhanced clinical awareness are needed to increase detection and reduce the impact of these neglected pathogens.

## Introduction

Rift Valley fever (RVF) virus (the causative agent of RVF) and *Coxiella burnetii* (the agent of Q fever) cause febrile illnesses that present significant public health challenges [1,2] and disproportionately affect sub-Saharan Africa [2,3]. These illnesses rank among the 13 global priority zoonoses but remain understudied, particularly in Eastern Africa, where they may have spread more widely [4,5].

Seasonal variations in temperature and rainfall significantly influence the spread of RVF virus and *C. burnetii* outbreaks in certain regions [6,7]. These changes increase vector populations, host susceptibility, and pathogen survival, ultimately increasing the risk to human health [7,8]. The RVF virus causes severe disease with

life-threatening complications, including muscle pain, vision loss, meningoencephalitis, hemorrhagic fever with hepatic involvement, renal failure, and, in severe cases, dysfunction of other vital organs [9]. East African countries bear the most significant burden, having experienced several outbreaks [10,11]. Studies have documented high seroprevalence in humans, even in the absence of outbreaks, in Sudan (21.5%) [12], Kenya (19.5%) [13], and Uganda (12.0%) [14]. To date, Ethiopia has reported no confirmed human cases of RVF, even though the country borders RVF endemic regions [11]. However, seroprevalence studies in the eastern (13.3%) and northeastern (6.1%) parts of the country have shown that healthy individuals in close contact with livestock have been exposed to RVF [15,16]. Ethiopia currently lacks vaccines for humans and animals [11].

*Coxiella burnetii* exist in two antigenic variants (Phase I and Phase II) [17], complicating clinical diagnosis because they cause a broad spectrum of disease manifestations [18,19]. Antibodies produced against these variants help distinguish between acute and chronic *C. burnetii*. Phase I IgG antibodies are induced against Phase I, indicating previous exposure or chronic *C. burnetii*, whereas Phase II IgM/ IgG antibodies produced against Phase II antigens are detected during acute/early infections [17,20–22]. This bacterium causes significant fever in several African countries. Studies have reported high seroprevalence rates in Egypt (25.1%) [23], Tanzania (20.3%) [24], where it surpasses malaria as a cause of fever, and Kenya (19.1%) [20], demonstrating its widespread presence across the African continent. Although limited community-based studies in Ethiopia have detected *C. burnetii* antibodies in healthy individuals [15,16], no study has determined its prevalence among febrile patients.

Historically, malaria has caused the majority of fevers in Ethiopia, but effective control strategies have significantly reduced the number of malaria cases [25]. As a result, many febrile patients currently test negative for malaria [26], suggesting that the RVF virus and *C. burnetii* could contribute to febrile illnesses. Limited research and diagnostic challenges have contributed to the scarcity of data on these infections. This study investigated the prevalence of RVF virus and *C. burnetii* infections among adult febrile patients visiting public health facilities in Eastern Ethiopia. The findings advance our understanding of these infections and their risk factors in Ethiopia, providing critical evidence to strengthen public health preparedness and response.

## Methods

### Ethics statement

This study was conducted in accordance with the Declaration of Helsinki. The Institutional Health Research Ethics Review Committee of the College of Health and Medical Sciences, Haramaya University (reference number: IHRERC/085/2022), and the Research Ethics Committee of the All Africa Leprosy Rehabilitation and Training Center/ Armauer Hansen Research Institute (reference number: PO-30–22) approved the study. All recruited febrile patients provided written, informed, and voluntary consent to participate in the study.

### Study settings

This study was conducted in the Shinile district of the Somali Region and Dire Dawa Administration, Eastern Ethiopia (Fig 1). The study areas were purposively selected based on accessibility, comparable climatic conditions, insufficient public health facilities, and limited disease surveillance systems. More than half of the residents (61.4%) in both districts lived in rural areas and received primary healthcare through 17 public health facilities (3 health facilities in Shinile and 14 health centers in Dire Dawa).

### Study design and population

A multisite public health facility-based cross-sectional study was conducted between March 01, 2023 - February 28, 2024. Data were collected year-round to ensure balanced febrile patient recruitment, account for seasonal and climatic variations among febrile case load and maintain proportional representation relative to the overall population.

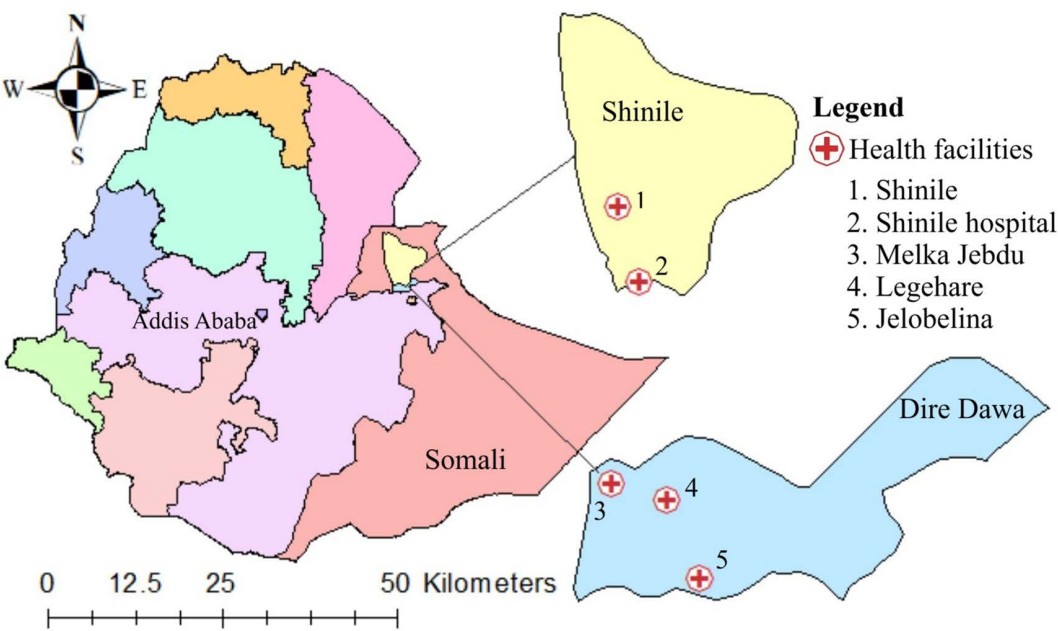

**Fig 1. Map of the Shinile district and Dire Dawa Administration displaying sampled health facilities.** *The map was generated using ArcGIS 10.8.2 (ESRI), integrating administrative boundaries from GADM (https://gadm.org) and study-collected health facility geodata, with all datasets available under a CC BY 4.0 license (https://creativecommons.org/).*

All febrile patients aged ≥18 years, who were residents of the study area, visited the selected health facilities, and were clinically diagnosed with sudden onset fever, were enrolled in the study. Fever was defined as an axillary temperature exceeding 37.5°C, accompanied by at least one of the following symptoms within the past 7 days: headache, muscle pain, chills, weakness, nausea, or any sign of bleeding. Febrile patients were excluded from the study if they tested positive for malaria, had a known infection, underwent surgery within 30 days, or received antibiotic treatment within two weeks before sampling.

## Sample size estimation

The sample size was determined using the Cochran single population proportion formula [27] assuming a standard score of 1.96 corresponding to a 95% confidence interval (CI) and a margin of error of ± 4% between the study and source populations. Seroprevalence rates of 19.5% for RVF and 19.1% for Q fever among febrile patients were obtained from studies conducted in Kenya [13,20]. The final sample size, adjusted for 10% non-response and confounding factors, was 415.

## Sampling techniques

Among the three public health facilities in the Shinile district, the Shinile health center and Shinile hospital were selected randomly (lottery method). This involved listing the names of all three public health facilities in Shinile on separate, identical slips of paper. The slips were placed in an opaque container and randomly drawn to ensure an unbiased selection. Similarly, Melka Jebdu, Legehare, and Jelobelina health centers were chosen from 14 health centers in the Dire Dawa Administration. The sample size was allocated proportionally based on the febrile patient load at each health facility (Fig 2). Febrile participants were randomly recruited using a predetermined interval of 2 or 3 among febrile patients presenting at each health facility. If a contacted febrile patient declined to participate, the next febrile patient was approached.

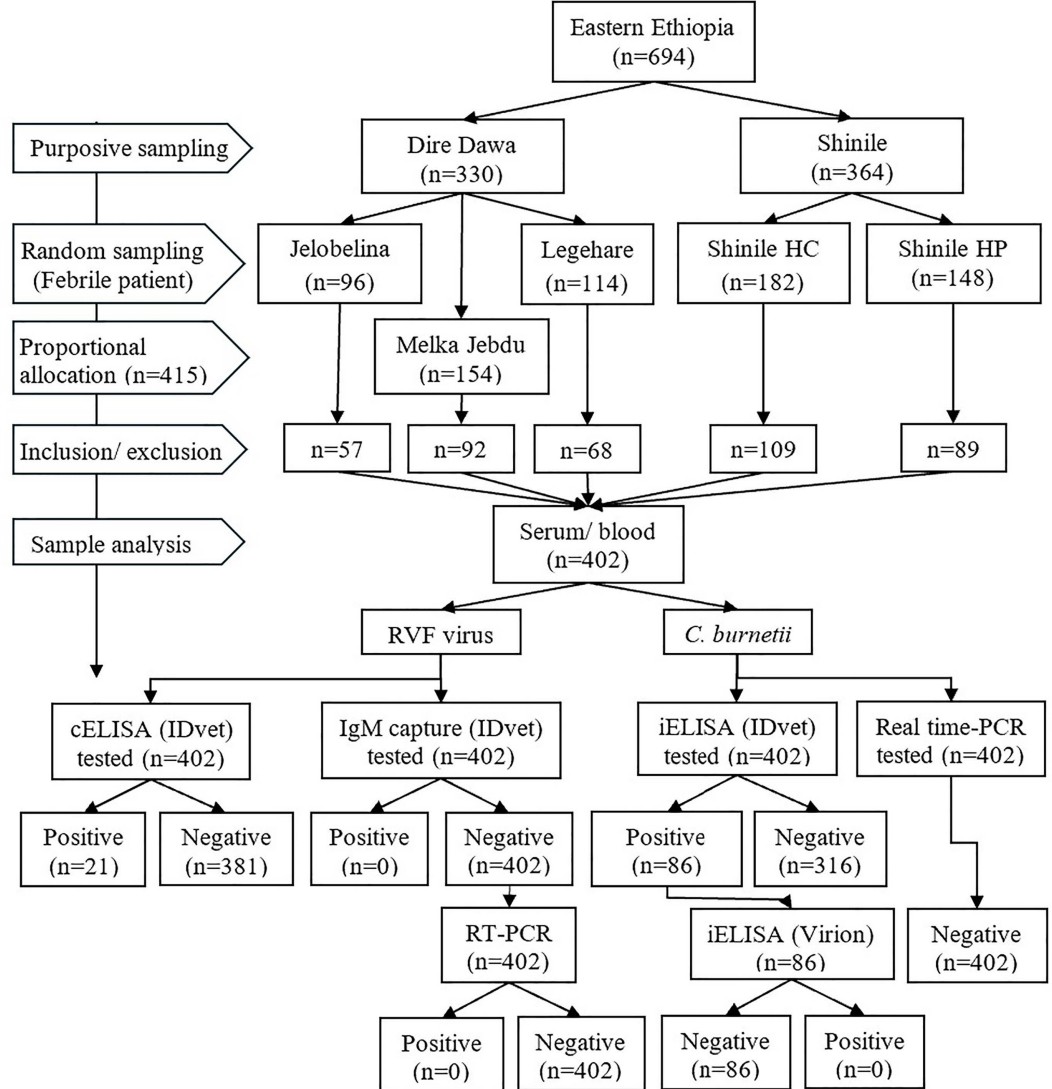

**Fig 2.** Flowchart of the study design and sample analysis. n: sample size; iELISA: indirect ELISA; cELISA: competitive ELISA; HP: hospital; HC: health center; HH: household.

## Data and sample collection

Data and blood samples were collected after obtaining written informed consent from the participants. Healthcare professionals screened the participants for eligibility, measured axillary temperature with a digital clinical thermometer, and collected clinical symptoms, sociodemographic data, and exposure history using structured questionnaires. The questionnaire was translated into the local languages by experts. A pretest of the study procedures (such as questionnaire, interview guides, and laboratory testing methods) was conducted with 5% of febrile patients at Jeldessa health center and Dilchora hospital, Dire Dawa, Ethiopia before the main study was conducted. Problems identified during the pre-test were addressed, necessary resources were secured, and vague or unclear questions were corrected.

Under aseptic conditions, 5 mL of venous blood was collected. Of this, 1 mL was aliquoted into BD vacutainer $K_2E$ (EDTA) tubes (Becton Dickinson Plymouth, UK; Reff: 367525), and then transferred to 2 mL microtube (SARSTEDT AG & Co. KG,

Germany; Reff: 72.694.005). EDTA blood samples were kept at -80°C until tested for detection of RVF virus using Reverse Transcription Polymerase Chain Reaction (RT-PCR), and *C. burnetii* using real-time PCR. The remaining 4 mL was aliquoted into a 6 mL BD trace element serum (Becton, Dickinson and Company, Franklin Lakes, New Jersey, USA; Reff: 368380) for Enzyme Linked Immunosorbent Assays (ELISA). The tube was centrifuged using a Hettich Universal 320 centrifuge (Andreas Hettich GmbH & Co. KG Tuttlingen, Germany) at 2500×g for 10 min. Up to 1.5 mL of serum was aliquoted into microtubes and stored at -20°C. All tests were conducted at the Hararghe Health Research Partnership Laboratory, Harar, Ethiopia.

## Laboratory assays

### Blood film examination

Thick and thin blood smears were prepared, Giemsa-stained, and examined at the respective health facility laboratories for *Plasmodium* species detection; however, parasite grading was not performed at enrollment.

### RVF virus antibody screening

The samples were screened simultaneously for RVF virus IgG, IgM, and viral RNA using multiple diagnostic methods, in accordance with the manufacturer's instructions.

Sera were analyzed using a competitive ELISA (IDvet Innovative Diagnostics, Louis Pasteur, Grabels, France) for detection of IgG against RVF virus nucleoprotein (NP), indicating previous exposure to RVF virus. Briefly, the sera, reagent, negative control (NC), and positive control (PC) were brought to room temperature. The wash buffer (20X) and anti-RVF-NP-PO conjugate (10X) were diluted to prepare a 1X working solution. The dilution buffer standardizes the sample viscosity for uniform microplate loading and minimizes nonspecific background interference. A 1:1 ratio mixture was prepared in a blank 96-well plate, containing 50 μL each of (1) dilution buffer, (2) test serum, (3) negative control (NC), and (4) positive control (PC) per well. This preparation protocol maintained the optimal sensitivity and specificity of the kit. The mixture was then transferred to antigen-precoated ELISA microplates and incubated at 37°C for 1 h. After washing with 300 μL of wash buffer using a BioTek 80TS washer (Agilent Technologies, Santa Clara, CA, USA), 100 μL of anti-RVF-NP-PO conjugate was added to each well. The plate was incubated at 22°C for 30 min, followed by another wash step. Next, 100 μL of substrate was added and incubated for 15 min, after which the reaction was stopped by adding 100 μL stop solution. Absorbance was measured at 450 nm using a BioTek 800TS Absorbance Reader (Agilent Technologies, Santa Clara, CA, USA). The competition percentage for each sample (S/N%) was determined by dividing its optical density (OD) ($OD_{sample}$) by that of the negative control ($OD_{NC}$) and multiplying by 100. Samples were classified as positive if the S/N% value was below 40%, negative if it exceeded 50%, and inconclusive if it fell between 40% and 50%. The manufacturer reported that the kit has a sensitivity of >91% and a specificity of 100%.

All sera were also screened using IgM capture ELISA (IDvet Innovative Diagnostics, Louis Pasteur, Grabels, France) to test for recent exposure. The procedure is the same as those for the IDvet Competitive ELISA, with minor differences. In the IgM capture ELISA, the RVF virus NP was added to the even-numbered columns of the well plate, and dilution buffer was added to the odd-numbered columns. The results were interpreted as per described [15]. According to the manufacturer, this kit has a diagnostic sensitivity of 98% and specificity of 100%.

### *C. burnetii* antibody screening

For the detection of *C. burnetii*, serum samples were analyzed in two steps: First, an initial screening for total (Phase I and Phase II) antibodies was performed using an indirect ELISA (IDvet Innovative Diagnostics, Grabels, France). This was followed by screening for Phase II IgG antibodies, also using indirect ELISA (Virion/Serion, Würzburg, Germany), and a *C. burneti* DNA genome screening, all according to the manufacturer's instructions.

Total antibodies were screened using IDvet indirect ELISA as follows: 5 μL each of NC, PC, and test serum were added to 245 μL of dilution buffer in a 96-well plate, resulting in a final dilution of 1:50. Next, 100 μL of pre-diluted NC and PCs

were transferred in duplicate to the *C. burnetii* antigen-coated plate, while test samples were dispensed as single replicates per well. The plate was washed, incubated, and OD readings were taken as previously described for competitive ELISA. The results were interpreted as previously described [16]. The manufacturer reported a sensitivity of 99% and a specificity of 98%.

Sera positive for total antibodies were further analyzed using an indirect ELISA (Virion/Serion, Würzburg, Germany) to screen for Phase II IgG antibodies, which indicate recent infection, as described previously [28]. Inconclusive results were reclassified as negative after retesting. The manufacturer states that the kit has 92.5% sensitivity and >99% specificity for Phase II IgG antibody. All ELISA tests were performed using the Agilent ELISA instrument mentioned above.

## Extraction and PCR

All blood samples were extracted using the QIAamp Viral RNA Mini Kit and QIAamp DNA Mini Kit (QIAGEN, Hilden, Germany), following the manufacturer's instructions. Nucleic acid quality was assessed using a UV5 spectrophotometer (Mettler-Toledo International Inc., Columbus, OH, USA) before individual testing for RVF virus and *C. burnetii* genomes.

A mixture of qScript XLT one-step RT-qPCR ToughMix Low Rox reaction (Quantabio, Massachusetts, USA), primers, and probes (Applied Biosystems, Waltham, MA, USA) was prepared according to the manufacturer's instructions. The primers and probes targeted a conserved domain in the RVF virus L-segment and the multicopy IS1111 element (7–120 copies) of *C. burnetii* (Table 1). Amplification was conducted using a QuantStudio 7 Flex Real-Time PCR System (Applied Biosystems, USA) under the following conditions: initial denaturation at 45°C for 10 min, followed by 40 cycles of denaturation at 94°C for 10 min, annealing and extension at 94°C for 30 s, and final extension at 60°C for 1 min. Samples were considered positive for both infections if amplification occurred within 36 cycles (Ct ≤ 36) [20,29].

## Quality control

ELISA results were verified by re-testing 5% of seropositive and seronegative samples using alternative methods: indirect human RVF virus ELISA (Abbexa) [30] and Q fever antibody ELISA (IDEXX, Liebefeld, Switzerland), both targeting IgG antibodies [28]. PCR results were also checked using the commercial Real Star RVF Virus RT-PCR (Altona Diagnostics GmbH, Hamburg, Germany) [31], and the AmpliSens *C. burnetii* FRT PCR Kit [32]. No discrepancies were observed between the primary and supplementary ELISA and PCR results.

## Data management and analysis

Data were entered into Epi Info 7.2.6.0, cleaned, and analyzed using STATA 17 (Stata Corp LLC, College Station, Texas, USA).

**Table 1. Description of primers and probes.**

| Primer/ probe name | Sequence (5′ to 3′) | Reference |
|---|---|---|
| RVF Forward | TGAAAATTCCTGAGACACATGG | [33] |
| RVF Reverse | ACTTCCTTGCATCATCTGATG | |
| RVF-probe | FAM: CACAAGTCCACACAGGCCCCTTACAT- BHQ1* | |
| *C. burnetii* Forward | CCGATCATTTGGGCGCT | |
| *C. burnetii* Reverse | CGGCGGTGTTTAGGC | |
| *C. burnetii* Probe | FAM: TTAACACGCCAAGAAACGTATCGCTGTG-BHQ1* | |

Note: *Black Hole Quencher-1

Fever severity was classified as mild (37.2–38.0°C), moderate (38.1–39.9°C), or high (>39.9°C). Symptom duration was categorized as <8, 8–14, or >14 days, based on average seroconversion times (6 days for RVF virus and 7 days for *C. burnetii*). Acute RVF infection was defined as IgM positivity or viral RNA detection. Acute *C. burnetii* infection was defined as the presence of Phase II IgG antibodies (with or without Phase I IgG), a higher Phase II IgG titer than Phase I IgG, or detection of Coxiella DNA. *C. burnetii* infection was defined as a Phase I IgG titer <800, while chronic infection was defined as a Phase I IgG titer ≥800 or a Phase I IgG titer exceeding the Phase II IgG titer. This classification accounts for antibody development 10–14 days after symptom onset, as recommended by the manufacturer and supported by previous studies [20,21]. The climatic seasons were classified into two main categories: rainy (April–September) and dry (October–March) season.

The variables were summarized using frequencies, means, standard deviations (SD), and ranges. Differences between study variables were assessed using Pearson's Chi-squared test ($\chi^2$). Bivariable and multivariable logistic regression analyses (with stepwise backward elimination, $p < 0.25$) were performed to calculate odds ratios (ORs) with 95% CIs. Age, a biologically important variable, was included in the multivariable model. Multicollinearity was evaluated using the variance inflation factor (VIF), and values below 10 were retained in the analysis. The model fit was evaluated using the Hosmer-Lemeshow test ($p > 0.05$). Statistical significance was set at $p < 0.05$.

## Results

### Demographic, and exposure characteristics

Sociodemographic characteristics, risk factors, clinical data, and venous blood samples were collected from 415 participants. Thirteen (0.3%) data were excluded due to missing questionnaires, insufficient samples, or duplicate study identification. This resulted in a final data analysis of 402 participants.

Of the febrile patients interviewed, 240 (59.7%) were male. The mean age was 31.1 (± 9.4 SD) years, with an age range of 18–70 years. More than half 215 (53.5%) were residing in urban settings and 278 (69.2%) were currently married.

Two hundred seventy-nine (69.4%) febrile patients reported having at least one domestic animal, including cattle, camels, sheep, goats, or a combination thereof. Since last month, a total of 34 (8.5%) febrile patients reported exposure to dead animal tissues, skin, abortion materials, or fluids, 40 (10.0%) were involved in animal slaughtering or meat processing, 73 (18.2%) helped with animal births or handled afterbirth materials, 106 (26.4%) slept near animal pens, 99 (24.6%) worked with animal manure, 103 (25.6%) engaged in milking or consuming raw milk, 56 (13.9%) assisted with animal feeding, and 51 (12.7%) cared for sick animals. More than half of the 238 (59.2%) febrile patients reported being bitten by a mosquito (Table 2).

### RVF virus, and *C. burnetii* infections

Overall, 21 (5.2%; 95% CI: 3.0–7.4) serum samples were IgG positive for RVF virus infection. A higher proportion of seropositive febrile patients was observed at the Shinile health center 8 (38.1%). None of the samples tested positive for RVF virus IgM antibodies.

A total of 86 (21.4%; 95% CI: 17.4–25.4) serum samples tested positive for IgG antibodies against Phases I and Phase II of *C. burnetii*. Among these, 58 (67.4%) exhibited Phase I IgG titers below 800, indicating a previous exposure. Nearly one-fourth of the samples 22 (25.6%) showed elevated anti-Phase I IgG levels (>800 titers; Phase I > Phase II), suggesting chronic infection. The remaining 6 (7.0%) had dominant Phase II IgG antibodies (Phase II > Phase I), indicating acute *C. burnetii* infection. Febrile patients at the Shinile health center had a higher exposure rate to *C. burnetii* 26 (24.5%) (Table 3).

**Table 2. Sociodemographic factors and exposure to RVF virus and *C. burnetii* among febrile patients.**

| Characteristics | | n(%) |
|---|---|---|
| Age group (in years) | 18-34 | 269(66.9) |
| | >34 | 133(33.1) |
| Educational status | Secondary and above* | 113(28.1) |
| | Primary | 126(31.3) |
| | No formal education | 163(40.6) |
| Occupation | Others** | 292(72.6) |
| | Farmers | 110(27.4) |
| Contact with ill family member | No | 380(94.5) |
| | Yes | 22(5.5) |
| Eaten wild animal meat | No | 354(88.1) |
| | Yes | 48(11.9) |
| PPE use during animal contact | No | 399(99.3) |
| | Yes | 3(0.7) |
| Travel history | No | 389(96.8) |
| | Yes | 13(3.2) |
| Mosquito abundance (<30 days) | No | 207(51.5) |
| | Yes | 195(48.5) |
| Use mosquito net since last month | No | 318(79.1) |
| | Yes | 84(20.9) |
| Spraying insecticides since last month | No | 378(94.0) |
| | Yes | 24(6.0) |
| Heard about RVF infection | No | 398(99.0) |
| | Yes | 4(1.0) |
| Heard about Q fever | No | 400(99.5) |
| | Yes | 2(0.5) |
| Seasonal variation | Dry | 260(64.7) |
| | Rainy | 142(35.3) |

n: sample size; *Grade 9 and above; **includes pastoralist, agropastoral, employee, housewife, merchant, student, and no job; PPE: personal protective equipment

### RNA and DNA detection

All blood samples tested for RVF virus using RT-PCR yielded negative results. Similarly, all samples, including the 6 positives for Phase II IgG antibodies, were negative for *C. burnetii* genomic DNA in the real-time PCR. However, three samples exhibited low levels of amplification, with Ct values between 35 and 36.8. These were classified as inconclusive in the primary testing but subsequently tested negative in the confirmatory test. All PCR controls confirmed the proper functioning of PCR detection.

### Clinical characteristics

The mean duration of illness before seeking healthcare consultation was $9.5 \pm 0.2$ days. None of the febrile patients exhibited typical symptoms of RVF virus infection, such as mucosal bleeding, bloody vomit, or epistaxis. For *C. burnetii* infection, a significant difference was observed among febrile patients with duration of symptoms ($p = 0.02$), and among those with diarrhea ($p = 0.001$) (Table 4).

**Table 3. RVF virus and *C. burnetii* seropositivity in health facilities.**

| Health facilities | n(%) | RVF virus IgG positive | *C. burnetii* IgG Phase I/ II | | | |
|---|---|---|---|---|---|---|
| | | | +/+(%) | PI < 800 | PI ≥ 800 | PII > PI |
| Jelobelina | 55(13.7) | 3 | 13(23.6) | 4 | 1 | 0 |
| Legehare | 65(16.2) | 2 | 9(13.9) | 11 | 6 | 1 |
| Melka Jebdu | 91(22.6) | 4 | 18(19.8) | 15 | 3 | 1 |
| Shinile | 106(26.4) | 8 | 26(24.5) | 21 | 8 | 3 |
| Shinile hospital | 85(21.2) | 4 | 20(23.5) | 7 | 4 | 1 |
| Total | | 21(5.2) | 86(21.4) | 58(67.4) | 22(25.6) | 6(7.0) |

n: sample size; PI: Phase I; PII: Phase II

**Table 4. Chi-square test of symptoms reported by febrile patients.**

| Signs and symptoms | | Frequency (%) | RVF virus IgG | χ² | *C. burnetii* IgG | χ² |
|---|---|---|---|---|---|---|
| | | | Pos (%) | | Pos (%) | |
| Duration of symptoms (days) | <8 | 151(37.6) | 8(5.3) | NA | 22(14.6) | p = 0.02 |
| | 8-14 | 187(46.5) | 11(5.9) | | 44(23.5) | |
| | >14 | 64(15.9) | 2(3.1) | | 20(31.3) | |
| Fever severity | Mild | 278(69.2) | 17(6.1) | NA | 58(20.7) | p = 0.23 |
| | Moderate | 115(28.6) | 4(3.5) | | 24(20.9) | |
| | High | 9(2.2) | 0(0.0) | | 4(44.4) | |
| Cough | Yes | 145(36.1) | 5(3.5) | p = 0.23 | 26(17.9) | p = 0.20 |
| | No | 257(63.9) | 16(6.2) | | 60(23.4) | |
| Sweating | Yes | 158(39.3) | 7(4.4) | p = 0.57 | 28(17.7) | p = 0.15 |
| | No | 244(60.7) | 14(5.7) | | 58(23.8) | |
| Abdominal pain | Yes | 74(18.4) | 3(4.1) | NA | 15(20.3) | p = 0.79 |
| | No | 328(81.6) | 18(5.5) | | 71(21.7) | |
| Joint/muscle aches | Yes | 298(74.1) | 16(5.4) | p = 0.83 | 57(19.1) | p = 0.06 |
| | No | 104(25.8) | 5(4.8) | | 29(27.9) | |
| Nausea | Yes | 60(14.9) | 5(8.3) | p = 0.24 | 13(21.7) | p = 0.96 |
| | No | 342(85.1) | 16(4.7) | | 73(21.4 | |
| Vomiting | Yes | 53(13.2) | 1(1.9) | NA | 14(26.4) | p = 0.34 |
| | No | 349(86.8) | 20(5.7) | | 72(20.6) | |
| Diarrhea | Yes | 29(7.2) | 1(3.5) | NA | 13(44.8) | p = 0.001 |
| | No | 373(92.8) | 20(5.4) | | 73(19.6) | |
| Loss of appetite | Yes | 178(44.3) | 12(6.7) | p = 0.22 | 43(24.2) | p = 0.23 |
| | No | 224(55.7) | 9(4.0) | | 43(19.2) | |

NA, not applicable; Pos: positive; χ²: chi-square

Healthcare professionals did not suspect RVF virus or *C. burnetii* infections in any of the selected febrile patients. Diagnoses based on clinical symptoms and history were classified as follows: 210 (52.1%) febrile patients with fever of unknown origin, 80 (22.1%) with presumptive malaria, 45 (11.2%) with acute respiratory infections, 26 (6.5%) with enteric fever, and 42 (10.4%) with other conditions (such as pneumonia, influenza, and chronic diseases). Various antibiotic panels have been used to treat these conditions.

## Factors associated with RVF virus infections

Bivariable logistic regression analysis revealed that febrile patients aged ≥35 years had significantly higher odds of RVF virus infection (odds ratio [OR]: 2.9, 95% CI: 1.2–7.0). In the multivariable logistic regression analysis, after adjusting for confounders, febrile patients aged ≥35 years had three times higher odds of RVF virus infection (adjusted OR [AOR]: 3.1, 95% CI: 1.3–7.8) than those aged <35 years (Table 5).

## Factors associated with *C. burnetii* infections

In bivariable logistic regression, the following factors were statistically associated with *C. burnetii* infection: being female (OR: 1.8, 95% CI: 1.1–2.8), rural residence (OR: 1.9, 95% CI: 1.2–3.1), and disposing of dead animals (OR: 2.5, 95% CI: 1.2–5.2). In the multivariable logistic regression model, females (AOR: 1.7, 95% CI: 1.1–2.9), rural residents (AOR: 2.4, 95% CI: 1.3–4.5), and those who disposed of dead animals (AOR: 2.6, 95% CI: 1.2–5.6) had higher odds of *C. burnetii* infection than their counterparts (Table 6).

**Table 5. Bivariable and multivariable analyses of predictors of RVF virus infection among febrile patients.**

| Characteristics | | n | RVF virus IgG Pos (%) | Crude OR (95% CI) | Adjusted OR (95% CI) | *p*-value |
|---|---|---|---|---|---|---|
| Sex | Male | 240 | 11(4.6) | 1.0 | 1.0 | 0.22 |
| | Female | 162 | 10(6.2) | 1.4(0.6-3.3) | 1.8(0.7-4.4) | |
| Age group (in years) | 18-34 | 269 | 9(3.4) | 1.0 | 1.0 | 0.01 |
| | ≥35 | 133 | 12(9.0) | 2.9(1.2-7.0) | 3.1(1.3-7.8) | |
| Seasonal variation | Dry | 257 | 11(4.2) | 1.0 | 1.0 | 0.17 |
| | Rainy | 145 | 10(7.0) | 1.7(0.7-4.1) | 1.9(0.7-4.7) | |

n: sample size; Pos, positive; Odds ratio; CI, confidence interval

**Table 6. Bivariable and multivariable analyses of predictors of *C. burnetii* infection among febrile patients.**

| Characteristics | | n | *C. burnetii* IgG Pos (%) | Crude OR (95% CI) | Adjusted OR (95% CI) | *P*-value |
|---|---|---|---|---|---|---|
| Sex | Male | 240 | 42(17.5) | 1.0 | 1.0 | 0.04 |
| | Female | 162 | 44(27.2) | 1.8(1.1-2.8) | 1.7(1.1- 2.9) | |
| Residence | Urban | 215 | 35(16.3) | 1.0 | 1.0 | 0.01 |
| | Rural | 187 | 51(27.3) | 1.9(1.2-3.1) | 2.4(1.3-4.5) | |
| Occupation | Others* | 292 | 66(22.6) | 1.0 | 1.0 | 0.10 |
| | Farmers | 110 | 20(18.2) | 0.8(0.4-1.3) | 0.6(0.3-1.1) | |
| Have domestic animals | No | 123 | 19(15.5) | 1.0 | 1.0 | 0.2 |
| | Yes | 279 | 67(24.0) | 1.7(0.8-3.0) | 1.5(0.8-2.9) | |
| Milking/ drinking milk | No | 299 | 65(21.7) | 1.0 | 1.0 | 0.2 |
| | Yes | 103 | 21(20.4) | 0.9(0.5-1.6) | 0.6(0.3-1.4) | |
| Disposing of dead animal | No | 368 | 73(19.8) | 1.0 | 1.0 | 0.02 |
| | Yes | 34 | 13(38.4) | 2.5(1.2-5.2) | 2.6(1.2-5.6) | |

n: sample size; Pos: positive; Odds ratio; CI: confidence interval; *includes agropastoral, housewives, traders, students, no job, butchers, day laborers, shop owners, drivers, etc.

## Discussion

The study identified a 5.2% (95% CI: 3.0–7.4%) seroprevalence of RVF virus among febrile patients, with no detectable IgM or viral RNA. The observed seroprevalence was markedly lower than rates reported in African countries, including Nigeria (21.5%) [34], Kenya (19.5%) [13], and Gabon (14.3%) [35]. The absence of acute-phase markers is likely due to delayed health-seeking behavior (mean: 9.5 days after symptom onset), consistent with findings from Rwanda [36]. Given that IgM and viral RNA are typically detectable only within the first 5–6 days post-infection [4], this delay probably precludes acute phase detection, as patients had already seroconverted to IgG while viral load dropped below PCR detection limits [37]. The lower seroprevalence and absence of acute infection in our study should not be interpreted as indicating a reduced risk but rather highlight the need for enhanced active surveillance. In resource-limited clinical settings, integrating point-of-care (POC) IgM testing with routine clinical evaluation could improve early case detection and facilitate real-time detection of outbreaks.

This study revealed that febrile patients aged ≥35 years had three times the odds of RVF virus exposure compared to younger febrile patients, a finding consistent with a previous study that demonstrated an age-associated risk pattern [35]. The increased susceptibility of older patients may be attributed to several factors. These include greater cumulative exposure to the RVF virus over their longer lifetime [20], age-related immunological decline associated with comorbidities, and malnutrition that may predispose to more severe infection and poorer health outcomes [38]. These observations have significant implications for understanding RVF epidemiology and developing targeted control strategies, particularly in endemic regions with aging populations. The relatively low seroprevalence rate of RVF virus infection in our study population limited the statistical power to comprehensively evaluate other potential risk factors, highlighting the need for larger-scale investigations to confirm this association and explore additional determinants of infection.

This study revealed a *C. burnetii* infection seroprevalence of 21.4% (95% CI: 17.4–25.4%), comparable to rates in Tanzania (20.3%) and Kenya (19.1%) [20] but lower than Egypt (55.0%) [39]. This study also identified a 7.0% seroprevalence of acute *C. burnetii* infection without detectable DNA. This discrepancy likely reflects the characteristically low bacterial load in blood samples (contrasting with higher yields from tissues or aborted materials) [9,40], delayed febrile patient presentation post symptoms, and potential antibiotic treatment. These diagnostic challenges underscore the need for enhanced clinical awareness of Q fever manifestations, paired serological testing (acute/ convalescent) for definitive diagnosis [18], and integrated surveillance combining POC testing, PCR, and epidemiological data.

An association with higher odds of *C. burnetii* infection was observed in females, rural residents, and individuals with a history of handling or disposing of dead animal bodies and tissues. While these findings strongly support previous epidemiological studies establishing close animal contact as a primary risk factor for human infection [15,16,21], the absence of animal infection status data in this study limits definitive confirmation of the transmission pathways. These results underscore the importance of targeted public health education regarding *C. burnetii* transmission risks and prevention strategies, particularly for high-exposure groups.

In these study settings, *C. burnetii* infection was not clinically suspected in any of the seropositive febrile patients. This misdiagnosis potentially leads to three critical consequences: underestimation of true infection, inappropriate treatment approaches, and delayed public health responses. The results underscore the necessity for healthcare worker training on *C. burnetii* infection recognition and the development of integrated febrile illness screening strategies. Furthermore, the implementation of accessible laboratory testing combined with reliable POC assays would facilitate accurate clinical triage and timely rapid diagnosis in resource-limited settings like Ethiopia.

This study innovatively investigated two priority pathogens in randomly selected febrile patients over an extended period, utilizing multiple sensitive and specific ELISA and PCR diagnostic methods alongside quality control tests. However, several important limitations must be acknowledged: (1) antibody cross-reactivity from potentially antigenically similar pathogens may have led to misclassification of RVF virus and *C. burnetii* infections; (2) delays in care-seeking

behavior could influence test results, particularly for IgM and genome markers; (3) resource constraints prevented the use of immunofluorescence assays (the reference technique) for pathogen identification; and (4) despite limiting the recall time to <30 days and cross-validating reports with clinical records, the reliance on self-reported data for symptom histories and exposure risks may introduce recall bias, potentially affecting risk factor analyses. Finally, the study design did not incorporate infection source tracing through household-level follow-up of febrile patients positive for either or both infections for animal and environmental sampling, limiting our ability to fully characterize transmission dynamics. These limitations should be carefully considered when extrapolating the findings to other settings.

## Conclusions

This study provides compelling evidence that the RVF virus and *C. burnetii* are significant yet underrecognized causes of febrile illness in Eastern Ethiopia. The key predictive factors were female sex, older age, and rural residence. Delayed healthcare-seeking among febrile patients precluded detection of acute-phase markers. These findings underscore the need for health authorities at all levels to consider these infections in the differential diagnosis of febrile illnesses and to address identified exposure risks. Policymakers are also advised to implement measures to mitigate their spread. Furthermore, active surveillance and longitudinal studies targeting high-risk areas are recommended to clarify transmission dynamics.

## Acknowledgments

We sincerely thank Haramaya University and Armauer Hansen Research Institute for ethical approval and logistical support for the research work. We are grateful to the Shinile and Dire Dawa Health and Agriculture Offices for granting permission to conduct this study in their administration, and to all participants for their voluntary involvement. We also acknowledge Jigjiga One Health Initiative Project for supplying diagnostic kits, and the Hararghe Health Research for providing consumables and laboratory equipment. Special thanks to Dr. Ashenafi Gebregiorgis (Armauer Hansen Research Institute) for his unreserved guidance, Adamu Tayachew (Ethiopian Public Health Institute) for providing kits, and Daniel Demissie (Hararghe Health Research) for his expertise in molecular detection and result interpretation.

## Author contributions

**Conceptualization:** Dadi Marami, Adane Mihret, Nega Assefa, Alemseged Abdissa, Adargachew Mulu, Rea Tschopp.

**Data curation:** Dadi Marami, Mahlet Osman, Gizachew Gemechu, Jacob S Witherbee.

**Formal analysis:** Dadi Marami.

**Investigation:** Dadi Marami, Mahlet Osman, Gizachew Gemechu, Jacob S Witherbee.

**Methodology:** Dadi Marami, Adane Mihret, Nega Assefa, Alemseged Abdissa, Adargachew Mulu, Rea Tschopp.

**Project administration:** Dadi Marami, Adane Mihret, Nega Assefa, Alemseged Abdissa, Adargachew Mulu, Rea Tschopp.

**Resources:** Dadi Marami, Gizachew Gemechu.

**Supervision:** Dadi Marami, Adane Mihret, Nega Assefa, Alemseged Abdissa, Mahlet Osman, Gizachew Gemechu, Adargachew Mulu, Rea Tschopp.

**Validation:** Dadi Marami, Adane Mihret, Nega Assefa, Alemseged Abdissa, Adargachew Mulu, Rea Tschopp.

**Visualization:** Dadi Marami, Adane Mihret, Nega Assefa, Alemseged Abdissa, Adargachew Mulu, Rea Tschopp.

**Writing – original draft:** Dadi Marami.

**Writing – review & editing:** Dadi Marami, Adane Mihret, Nega Assefa, Alemseged Abdissa, Mahlet Osman, Gizachew Gemechu, Jacob S Witherbee, Adargachew Mulu, Rea Tschopp.

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
