## [Decision Letter · Decision Letter 0]

18 Jun 2025

Rift Valley fever virus and Coxiella burnetii infections among febrile patients, Eastern Ethiopia

Dear Dr. Marami,

Thank you for submitting your manuscript to PLOS Neglected Tropical Diseases. After careful consideration, we feel that it has merit but does not fully meet PLOS Neglected Tropical Diseases's publication criteria as it currently stands. Therefore, we invite you to submit a revised version of the manuscript that addresses the points raised during the review process.

Please submit your revised manuscript within 60 days Aug 17 2025 11:59PM. If you will need more time than this to complete your revisions, please reply to this message or contact the journal office at plosntds@plos.org. Please include the following items when submitting your revised manuscript:

We look forward to receiving your revised manuscript.

Kind regards,

Jonas Klingström

Academic Editor

Justin Remais

Section Editor

Shaden Kamhawi

co-Editor-in-Chief

Paul Brindley

co-Editor-in-Chief

**Journal Requirements:**

At this stage, the following Authors/Authors require contributions: Dadi Marami, Adane Mihret, Nega Assefa, Alemseged Abdissa, Mahlet Osman, Gizachew Gemechu, Jacob S Witherbee, Adargachew Mulu, and Rea Tschopp. Please ensure that the full contributions of each author are acknowledged in the "Add/Edit/Remove Authors" section of our submission form.

- ® on pages: 8, 9, and 11

- TM on pages: 9, 10, and 11.

3) Thank you for including an Ethics Statement for your study. Please state whether the consent obtained is verbal or written.

4) Please upload them main figure as a separate Figure file in .tif or .eps format. For more information about how to convert and format your figure files please see our guidelines: 

Potential Copyright Issues:

i) Figure 1. Please (a) provide a direct link to the base layer of the map (i.e., the country or region border shape) and ensure this is also included in the figure legend; and (b) provide a link to the terms of use / license information for the base layer image or shapefile. We cannot publish proprietary or copyrighted maps (e.g. Google Maps, Mapquest) and the terms of use for your map base layer must be compatible with our CC BY 4.0 license.

6) Please provide a completed 'Competing Interests' statement, including any COIs declared by your co-authors. If you have no competing interests to declare, please state "The authors have declared that no competing interests exist". 

**Reviewers' Comments:**

Reviewer's Responses to Questions

**Key Review Criteria Required for Acceptance?**

**Methods**

-Are the objectives of the study clearly articulated with a clear testable hypothesis stated?

-Is the study design appropriate to address the stated objectives?

-Is the population clearly described and appropriate for the hypothesis being tested?

-Is the sample size sufficient to ensure adequate power to address the hypothesis being tested?

-Were correct statistical analysis used to support conclusions?

-Are there concerns about ethical or regulatory requirements being met?

Reviewer #1: -Are the objectives of the study clearly articulated with a clear testable hypothesis stated? Yes

-Is the study design appropriate to address the stated objectives? Need a review

-Is the population clearly described and appropriate for the hypothesis being tested? No

-Is the sample size sufficient to ensure adequate power to address the hypothesis being tested? Yes

-Were correct statistical analysis used to support conclusions? Yes

-Are there concerns about ethical or regulatory requirements being met? No

Reviewer #2: The methods including study design are appropriate to answer the questions posed about seroprevalence in this population for RVF and Coxiella. One point of interest would be participation rate-i.e. how many patients agreed to participate or declined participation in the study? Also, was compensation offered to participants?

**Results**

-Does the analysis presented match the analysis plan?

-Are the results clearly and completely presented?

-Are the figures (Tables, Images) of sufficient quality for clarity?

Reviewer #1: -Does the analysis presented match the analysis plan? Not fully.

-Are the results clearly and completely presented? Partly.

-Are the figures (Tables, Images) of sufficient quality for clarity?Can be improved.

Reviewer #2: Results were clearly presented. In Table 2, I think it would be helpful for an international audience to provide an exchange rate for a Birr to US dollars at the time of the study to better assess socioeconomic status.

**Conclusions**

-Are the conclusions supported by the data presented?

-Are the limitations of analysis clearly described?

-Do the authors discuss how these data can be helpful to advance our understanding of the topic under study?

-Is public health relevance addressed?

Reviewer #1: -Are the conclusions supported by the data presented? Partly.

-Are the limitations of analysis clearly described? More or less

-Do the authors discuss how these data can be helpful to advance our understanding of the topic under study? More or less

-Is public health relevance addressed? Yes

Reviewer #2: Conclusions are clearly stated and limitations address inherent issues in this type of research.

**Editorial and Data Presentation Modifications?**

Reviewer #1: (No Response)

Reviewer #2: Please see above questions.

**Summary and General Comments**

Reviewer #1: Abstract

1) The way lines 20-21 are written: I think it should be IgM positive samples that should be sequenced as they are linked to an acute infection, whereas IgG samples should be associated with chronical or permanent infection…Can authors clarify this? Also the whole sentence does not make sense, because it is inaccurate “…IgG-positive samples were tested for genomic markers using real-time polymerase chain reaction”.

2) Can authors explain the meaning of this sentence, it looks confusing for me: “IgG-positive samples were tested for genomic markers using real-time polymerase chain reaction”.

3) Conclusion to be re-written.

Introduction

4) Use active form as much as possible while maintaining a classical structure of a sentence (Subject + Verb + Object) throughout the manuscript. For examples lines 108-100, 112-113, etc.

Methods

5) Since all symptoms were self-reported could authors include or address any bias related to that?

6) Lines 116-119: rephrase. Can the authors clarify how they objectivized these elements?

“Known infection”, “clinical evidence of a specific cause of fever”, “positivity to Malaria”?

7) Sample size estimation: Instead of putting the raw formula, just mention the name of the formula and the reference.

8) On which basis the prevalence of RVF Kenya was chosen, why not that from Egypt or Tanzania? Can authors lay more on that?

9) Lines 127-128: rephrase this sentence or check the punctuation. What authors mean by lottery method? Is there any reference for that, or authors can explain?

10) Lines 130-132: the recruitment method is blurred, not sure it’s clear for the reader. If fever was the main inclusion criterium, it means that only febrile patients were randomly selected, not all patients. In that case, the sentence should read as follows “Febrile participants were randomly recruited using a predetermined interval of 2 or 3 among them or among febrile patients…”. Otherwise, the way it is written, the interval of 2 or 3 correspond to all patients, not febrile ones. Authors can clarify.

11) Lines 133-134: authors can explain shortly the procedure. Were authors contacting febrile or all patients?

12) Line 140: specify how the pretest was carried out

13) Lines 141-143: not sure that tube as per the international codification…Check or provide the reference.

14) Lines 147-148: drop, not useful

15) Lines 144-145: real-time polymerase chain reaction corresponds to rt PCR, not RT-PCR which is Retro-Transcriptase polymerase chain reaction. Authors should choose what they used for the study. May I recall that we have here an RNA virus and a DNA organism (bacterium).

16) Lines 145-146, if authors talk about centrifugation, better for them to also mention the centrifuge brand.

17) Specify the ELISA instrument brand used for the RVFV runs.

18) Authors can specify the type of ELISA used for RVF screening, as they did for Q fever screening?

19) Lines 161-162 are not clear enough for the reader. Authors can explain the added value of RVFV and dilution buffer adds-on.

20) Lines 168-170 is not clear for me: please present the kit and its targets, then the manufacturer’s performance.

21) Lines 164-171 to be rephrased.

22) Lines 177-180: when samples were IgG positive, they went through a IgM screening as per the authors’ algorithm (lines 154-159, 205-206): why authors did not screen only IgM positive samples for more accuracy?

23) Lines 188-190: were the Ct-values thresholds set by the manufacturer or the authors?

24) Table 1 is for C. burnetii or for both pathogens? Please specify and adjust the table’s title.

25) Lines 194-196: can authors specify the target product profile for all ELISA assays IdVet, Abbexa, and IDEXX to justify the way the algorithm was set. Or authors can provide some arguments on why they should start with Idvet, then go for Abbexa and IDEXX for QC. Maybe a reference can be sufficient to illustrate this. Idem for PCR kits.

Results

26) Provide a flow diagram for lines 222-227. After this provide demographics, clinical results thereafter, go the test results.

27) Table 2, the title can be summarized.

28) Line 243: is RVF positivity referred to IgM positive samples?

29) If this assumption is true “None of the RVF virus IgG-positive patients tested positive for RVF virus IgM” (lines 244-245): why authors should sequence IgG positive samples?

30) Lines 257-259: if the IgG positive samples selected (RVF or Coxiella) for rt PCR did not include any IgM positive, I am not surprised to have negative signals in PCR. For this, the criteria of samples selection should be revised and the results of the study reconsidered.

31) Lines 263-266: are “mucosal bleeding”, “bloody vomit”, or “epistaxis” most common symptoms in classical clinical picture? Authors should check the frequency of these symptoms in common RVF infection.

32) Titles of tables are very long. Shorten!

33) RVF and Coxiella results are not clearly presented.

Discussion

34) Lines 302-303: would the fact of having IgM negative for both pathogens that led the authors to screen IgG positive samples through rt PCR? If yes, this should be point out somewhere in the methods to give a logical thread of testing in the study.

35) Lines 353-357: correctly number the points in the limitations section.

36) Diagnostic tools, cut-off for assays interpretation, the time elapsed between disease onset and the care seeking, samples collection, combining point-of-care to clinical picture in the ground conditions to quickly triage and isolated patients should be sufficiently discussed.

37) From all changes, a conclusion can be re-written to include any change.

Reviewer #2: Well-written paper that addresses a need in this geographic region. Primary prevention efforts regarding Coxiella exposure could be further discussed but realize this is challenging in this setting due to resource limitations.

PLOS authors have the option to publish the peer review history of their article (what does this mean? ). If published, this will include your full peer review and any attached files.

**Do you want your identity to be public for this peer review?** For information about this choice, including consent withdrawal, please see our Privacy Policy .

Reviewer #1: No

Reviewer #2: No

**Figure resubmission:**

**Reproducibility:**



---

## [Decision Letter · Decision Letter 1]

19 Jul 2025

Dear Mr Marami,

We are pleased to inform you that your manuscript 'Rift Valley fever virus and Coxiella burnetii infections among febrile patients, Eastern Ethiopia' has been provisionally accepted for publication in PLOS Neglected Tropical Diseases.

Best regards,

Jonas Klingström

Academic Editor

Justin Remais

Section Editor

Shaden Kamhawi

co-Editor-in-Chief

Paul Brindley

co-Editor-in-Chief

Reviewer's Responses to Questions

**Key Review Criteria Required for Acceptance?**

**Methods**

-Are the objectives of the study clearly articulated with a clear testable hypothesis stated?

-Is the study design appropriate to address the stated objectives?

-Is the population clearly described and appropriate for the hypothesis being tested?

-Is the sample size sufficient to ensure adequate power to address the hypothesis being tested?

-Were correct statistical analysis used to support conclusions?

-Are there concerns about ethical or regulatory requirements being met?

Reviewer #1: Most points addressed.

Reviewer #2: (No Response)

**Results**

-Does the analysis presented match the analysis plan?

-Are the results clearly and completely presented?

-Are the figures (Tables, Images) of sufficient quality for clarity?

Reviewer #1: Most points addressed.

Reviewer #2: (No Response)

**Conclusions**

-Are the conclusions supported by the data presented?

-Are the limitations of analysis clearly described?

-Do the authors discuss how these data can be helpful to advance our understanding of the topic under study?

-Is public health relevance addressed?

Reviewer #1: Most points addressed.

Reviewer #2: (No Response)

**Editorial and Data Presentation Modifications?**

Reviewer #1: (No Response)

Reviewer #2: (No Response)

**Summary and General Comments**

Reviewer #1: None

Reviewer #2: (No Response)

PLOS authors have the option to publish the peer review history of their article (what does this mean? ). If published, this will include your full peer review and any attached files.

**Do you want your identity to be public for this peer review?** For information about this choice, including consent withdrawal, please see our Privacy Policy .

Reviewer #1: No

Reviewer #2: No

---

## [Editor Report · Acceptance letter]

Dear Mr Marami,

We are delighted to inform you that your manuscript, " 

Rift Valley fever virus and Coxiella burnetii infections among febrile patients, Eastern Ethiopia," has been formally accepted for publication in PLOS Neglected Tropical Diseases.

Best regards,

Shaden Kamhawi

co-Editor-in-Chief

Paul Brindley

co-Editor-in-Chief
